# Personality traits and other factors associated with psychotropic medication non-adherence at two hospitals in Uganda. A cross-sectional study

**Emmanuel Niyokwizera**[1]*, **David Nitunga**[1], **Joshua Muhumuza**[2], **Raissa Marie Ingrid Niyubahwe**[1], **Nnaemeka Chukwudum Abamara**[1,3], **Joseph Kirabira**[1,4]

1 Department of Mental Health and Psychiatry, Kampala International University—Western Campus, Bushenyi, Uganda, 2 Department of Surgery, Kampala International University—Western Campus, Bushenyi, Uganda, 3 Department of Psychology, Nnamdi Azikiwe University, Awka, Anambra State, Nigeria, 4 Department of Psychiatry, Faculty of Health Sciences, Busitema University, Mbale, Uganda

* emmanuel.niyokwizera@studwc.kiu.ac.ug

**Data Availability Statement:** All relevant data are within the manuscript and its Supporting Information files.

## Abstract

Mental illnesses, like other chronic illnesses, require medications for both immediate, short term and long-term treatment. Medication adherence is the first and most important factor for better treatment outcomes. Non-adherence to psychotropic medications is associated with relapse, readmission, and early death. Psychological factors are among the common factors associated with non-adherence. Specific personality traits moderate the beliefs about medication that influence non-adherence to medications. Sociodemographic and clinical factors can also influence non-adherence to psychotropic medications. Non-adherence to psychotropic medications is high in Africa. Still, to the best of our knowledge, there is a lack of studies on the level of psychotropic medication non-adherence and associated personality traits. The aim was to determine the prevalence of psychotropic medication non-adherence and associated personality traits among people with mental illness attending Kampala International University Teaching Hospital (KIU-TH) and Jinja Regional Referral Hospital (JRRH). This study employed a hospital-based cross-sectional design. 396 adult patients suffering from mental illness were collected from KIU-TH and JRRH outpatient clinics. Medication adherence was assessed using the Medication Adherence Rating Scale (MARS) while personality traits were assessed by the short form of the Big Five Inventory (Ten Items Personality Inventory). In our study, we first assessed sociodemographic and clinical factors influencing psychotropic medication non-adherence (confounders). A questionnaire with sociodemographic information was also used. Logistic regression was used to assess personality traits and other factors associated with psychotropic medication non-adherence. The majority of the study participants were males (59.1%), from rural areas (74.2%), with a secondary educational level (47.5%) and unemployed (44.9%). The prevalence of psychotropic medication was 46.21%. Poor family support (aOR = 6.915, CI = 3.679–12.998, P<0.001), belief in witchcraft/sorcery (aOR = 2.959, CI = 1.488–5.884, P = 0.002), experiencing side effects (aOR = 2.257, CI = 1.326–3.843, P = 0.003), and substance use (aOR = 4.174, CI = 2.121–8.214, P<0.001) were factors significantly associated

**Funding:** The author(s) received no specific funding for this work.

**Competing interests:** The authors have declared that no competing interest exist.

**Abbreviations:** **JRRH**, Jinja Regional Referral Hospital; **KIU-TH**, Kampala International University Teaching Hospital; **MARS**, Medication Adherence Rating Scale; **TIPI**, Ten Items Personality Inventory.

with psychotropic medication non-adherence. The personality traits significantly associated with psychotropic medication non-adherence after controlling for the confounders were neuroticism (aOR = 7.424, CI = 3.890–14.168, P<0.001) and agreeableness (aOR = 0.062, CI = 0.024–0.160, P<0.001). In this study, medication non-adherence was high. Non-adherent patients were more likely to have predominant neuroticism personality traits. Non-adherence to medication was shown to be less common in individuals with agreeableness personality traits. Other factors associated with psychotropic medication non-adherence were poor social support, witchcraft beliefs, the presence of side effects, and substance use. Specific interventions should be done for patients with a high risk of being non-adherent to psychotropic medications, with the involvement of all stakeholders including caregivers, parents, tutors, and trustees.

## 1. Introduction

Worldwide, psychiatric disorders are among the common public health problems and are responsible for more economic burdens than somatic diseases like diabetes and cancer [1]. By 2050, the burden due to psychiatric disorders and substance use disorders is estimated to increase by 139% in Eastern Africa if there is no change in the prevalence and management of these disorders [2].

The 2017 report by WHO indicated that globally, 80% of people with mental disorders were living in low- and middle-income countries (LMICs) which include Uganda, and up to 75% of affected persons in many LMICs did not have access to the needed treatment [3]. In Uganda, the prevalence of mental disorders is 22, 9% in children and 24, 2% in adults, and around 80% of the population with psychiatric disorders uses both traditional and modern treatment which can predict poor adherence to medications [4].

Patients with psychiatric disorders are more likely to be non-adherent to treatment than those with other chronic medical diseases [1]. Worldwide, the prevalence of non-adherence to medication among psychiatric disorders is estimated to be 49% and factors associated with non-adherence vary in different countries [5]. In Africa, the prevalence of non-adherence is around 48% but varies between 20–80% [5–8]. Medication non-adherence is a complex and multifaceted healthcare issue. It is influenced by different factors divided into patient-related factors, physician-related factors, and health system/team-building-related factors [9].

Psycho-social and demographic factors (age, gender, marital status, occupation, family and social support, etc.), substance abuse, and beliefs about mental illness are important factors associated with non-adherence related to the patient [5]. The results of a study done in Ethiopia stated that non-adherence is associated with old age [10]. With aging, the frequency of chronic diseases, physical and cognitive deficiencies, and the number of medications used increase, which causes non-adherence to treatment as an important health problem [11]. During pregnancy, females have been shown to have a greater risk of experiencing adverse drug reactions as a result of sex differences in pharmacokinetics and pharmacodynamics which increase non-adherence [12, 13]. Unemployment combined with a lack of social support leads to poverty and thoughts of abandonment which predispose to non-adherence to medications [14]. In Uganda, traditional beliefs about mental disorders are still present in society and could influence medical treatment adherence [15]. Illness-related factors like side effects of medications, beliefs about the illness, treatment complexity, and lack of insight are barriers to medication adherence [16]. The beliefs about medication that influence non-adherence are

moderated by specific personality traits [17]. Personality traits can predict healthcare utilization as well as health outcomes [18]. In Italy, neuroticism and conscientiousness were the most common personality traits associated with psychotropic medication non-adherence. High levels of conscientiousness personality traits were related to poorer understanding of the clinical information received from healthcare professionals [19]. In India, conscientiousness and agreeableness personality traits were associated with medication adherence [20]. There is a gap in research about psychotropic medication non-adherence and associated personality traits in Africa (we didn't find any study in Africa to the best of our knowledge).

Non-adherence to psychiatric drugs is associated with relapse, re-hospitalization, and premature death [16]. Despite the free access to medications in some regions, non-adherence to psychotropic medications is still high in Eastern Africa but there is currently a paucity of published studies on the level of psychotropic medication non-adherence and associated factors in Uganda [21].

Assessment of medication adherence could help to understand the reasons for non-adherence in people with mental disorders and create the framework for therapies aimed at boosting adherence. Therefore, this study aimed to determine the prevalence of psychotropic medication non-adherence and associated personality traits (as well as other factors) in Uganda.

## 2. Methods

### 2.1 Aim, design, setting, and study population

We aimed to determine the prevalence of psychotropic medication non-adherence and associated personality traits among patients with mental illness. Before reaching that objective, we first assessed other possible factors (confounders) that could influence medication non-adherence in our study participants. A descriptive cross-sectional study employing quantitative methods was conducted from 1$^{st}$ July to 1$^{st}$ September 2023 among people with mental disorders attending psychiatric outpatient clinics at Kampala International University-Teaching Hospital and Jinja Regional Referral Hospital. Patients diagnosed with mental disorders and taking psychotropic medications for at least 6 months, aged 18 years old and above were included in the study. Patients with active psychiatric symptoms were excluded from the study.

### 2.2 Sample size estimation, recruitment, ethical considerations, and data collection process

The sample size was calculated using the single proportion formula for the present study (Daniel formula, 1995): n = Z$^2$ P (1-P)/d$^2$ which gave 360 participants. By adding 10% we found a total of 396 patients with mental illness who were recruited from outpatient clinics of Kampala International University-Teaching Hospital and Jinja Regional Referral Hospital. Patients were chosen using a simple random sampling method.

Patients who came for psychiatric review at outpatient clinics and fulfilled the inclusion criteria were approached and explained the purpose of the study.

A pilot study was conducted on 40 patients who were not part of the study participants. Approval to carry out the study was sought and obtained from the Research Ethics Committee of Bishop Stuart University (the research number is: BSU-REC-2023-99) and it has been conducted according to the principles expressed in the declaration of Helsinki. Before data collection, written informed consent from participants was obtained after fully explaining the details of the study to them in English and local Languages. Before asking for consent from the patients, we first assessed their mental capacity using the Functional test of capacity.

Participants were not forced to enroll themselves if they didn't want to. Participants were free to withdraw from the study any time he/she wished without coercion or compromise of care they were entitled to. As patients with mental illness are among the vulnerable population, written informed consent was also obtained from the next of kin and/or legally authorized representative of the patients before study participation.

Those who consented to participate in the study were administered a questionnaire by 3 psychiatrists and 1 psychiatric clinical officer. The primary investigator and research assistants collected the data. Throughout the data collection phase, research assistants got ongoing training and were supervised by the primary investigator. The questionnaire comprised sociodemographic information, clinical characteristics, the Medication Adherence Rating Scale (MARS), and the Ten Item Personality Inventory (TIPI) scale. Identification of participants was by means of codes. The socio-demographic information included age, gender, residence, level of education, marital status, employment status, and family support. The questionnaire included also the diagnosis of the patient (confirmed by clinical interview using DSM-V-TR diagnostic criteria), associated comorbidities, duration of illness and duration of continuous therapy, number of tablets taken by the patient per day, the frequency as well as side effects of medications.

Patients' medication adherence was assessed using the Medication Adherence Rating Scale (MARS). The MARS comprised 10 items with the response of yes or no and a total score of 10. A total score then revealed the level of adherence, scoring less than 6 was considered as non-adherence while 6 and above was considered as adherence [22]. To assess the personality traits, we used the Ten-Item Personality Inventory, which is a short version of the Big Five Inventory. It consists of ten pairs of adjectives generated using a 7-point Likert scale, with 7 (strongly agree) being the highest and 1 being the lowest (strongly disagree) [23]. TIPI scoring ("R" denotes reverse-scored items): Extraversion (1, 6R); Agreeableness (2R, 7); Conscientiousness (3, 8R); Neuroticism (4, 9R); Openness to Experiences (5, 10R). The score of the personality trait is the average of the 2 items. If the score of that personality trait is above the mean, the person scored high on that trait. The questionnaire and the tools were pre-tested before being used in the study.

## 2.3 Data processing and analysis

After being prepared and encoded, data were entered in Microsoft Excel version 2016 before being exported to Stata 15 for analysis. Proportions, percentages, and frequencies were calculated and presented in a table. Both bivariate and multivariate logistic regression analyses were used to assess personality traits and other factors associated with psychotropic medication non-adherence. For the multivariate analysis, variables that had a p-value below 0.2 at bivariate analysis were taken into consideration. The odds ratio (ORs) for the association were also presented, along with the appropriate 95% CI and p-values. Factors with a p-value less than 0.05 were significantly associated with non-adherence. The personality trait for which the p-value was less than 0.05 after controlling for the other possible factors was considered as significantly associated with non-adherence to psychiatric drugs.

## 3. Results

During the study period, to fulfill the expected sample size of 396 participants, we approached 410 patients with mental illness who came for review, but only 397 patients fulfilled the inclusion criteria. One did not consent to participate in the study. The 396 patients who consented to participate in the study had the questionnaire filled and adherence to medications assessed (Fig 1).

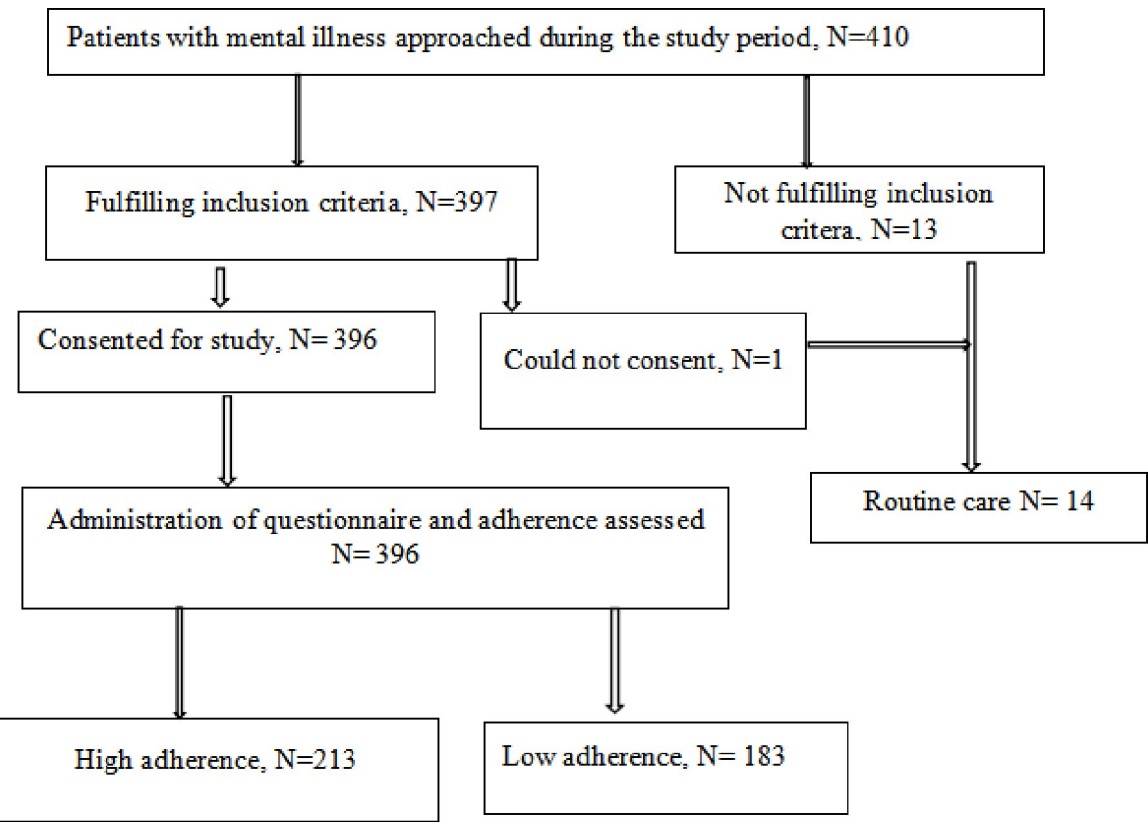

**Fig 1. Participant flowchart.**

### 3.1 Sociodemographic and clinical characteristics of study participants

Males made up 234 (59.1%) of the study participants, with the majority falling into the 18–30 (42.7%) and 31–45 (41.2%) age categories. The majority were from pastoral areas 294(74.2%) and had a mental illness for 1–5 years (71.2%). The other baseline characteristics are shown in Table 1 below.

### 3.2 Prevalence of psychotropic medication non-adherence among people with mental illness at KIU-TH and JRRH

In this study, 213 patients were adherent while 183 were non-adherent to psychotropic medication translating to 53.79% (95%CI 49.2–58.6) for adherence and 46.21(95%CI 41.4–50.8) for non-adherence respectively.

### 3.3 Sociodemographic and clinical factors associated with psychotropic medication non-adherence among people with mental illness at KIU-TH and JRRH

At the bivariate level of analysis, the variables that had a p-value less than 0.2, and therefore qualified for multivariate analysis were: sex, education level, employment status, marital status, family support, beliefs about mental illness, pill frequency, side effects, chronic disease, and substance abuse. The rest of the details are shown in Table 2 below.

**Table 1. Sociodemographic and clinical characteristics of study participants.**

| Characteristic | Overall N = 396 n (%) | Adherence, N = 213 n (%) | Non-adherence, N = 183 n (%) | P value |
|---|---|---|---|---|
| **Age category** | | | | 0.778 |
| 18–30 | 169(42.7) | 91(53.8) | 78(46.2) | |
| 31–45 | 163(41.2) | 90(55.2) | 73(44.8) | |
| 46–75 | 64(16.2) | 32(50.0) | 32(50.0) | |
| **Sex** | | | | 0.081 |
| Male | 234(59.1) | 117(50.0) | 117(50.0) | |
| Female | 162(40.9) | 96(59.2) | 66(40.8) | |
| **Residence** | | | | 0.818 |
| Rural | 294(74.2) | 157(53.4) | 137(46.6) | |
| Urban | 102(25.8) | 56(54.9) | 46(45.1) | |
| **Education level** | | | | **0.001** |
| None | 34(8.6) | 19(55.8) | 15(44.2) | |
| Primary | 109(27.5) | 44(40.4) | 65(59.6) | |
| Secondary | 188(47.5) | 104(55.3) | 84(44.7) | |
| Tertiary | 65(16.4) | 46(70.7) | 19(29.3) | |
| **Employment** | | | | 0.028 |
| Employed | 79(19.9) | 51(64.5) | 28(35.5) | |
| Self Employed | 139(35.1) | 64(46.0) | 75(54.0) | |
| Unemployed | 178(44.9) | 98(55.0) | 80(45.0) | |
| **Religion** | | | | 0.237 |
| Anglican | 130(32.8) | 68(52.3) | 62(47.7) | |
| Catholic | 68(17.1) | 36(52.9) | 32(47.3) | |
| Pentecostal churches | 110(27.8) | 56(50.9) | 54(49.1) | |
| Muslim | 83(21.0) | 48(57.8) | 35(42.2) | |
| None | 5(1.3) | 2(40.0) | 3(60.0) | |
| **Marital status** | | | | 0.042 |
| Married | 125(31.6) | 77(61.6) | 48(38.4) | |
| Single | 191(48.2) | 103(53.9) | 88(46.1) | |
| Separated/Divorced | 62(15.7) | 25(40.3) | 37(59.7) | |
| Widow | 18(4.5) | 8(44.4) | 10(55.6) | |
| **Family support** | | | | **<0.001** |
| Poor | 111(28.0) | 26(23.4) | 85(76.6) | |
| Good | 285(72.0) | 187(65.6) | 98(34.4) | |
| **Hospital** | | | | 0.362 |
| JRRH | 324(81.8) | 178(54.9) | 146(45.1) | |
| KIU-TH | 72(18.2) | 35(48.6) | 37(51.4) | |
| **Beliefs about mental illness** | | | | **<0.001** |
| Brain disorder | 255(64.4) | 159(62.3) | 96(37.7) | |
| Witchcraft/sorcery | 67(16.9) | 22(32.8) | 45(67.2) | |
| Demon possession | 38(9.6) | 19(50.0) | 19(50.0) | |
| Ancestor's spirit | 36(9.1) | 13(36.1) | 23(63.9) | |
| **Diagnosis** | | | | 0.211 |
| BAD | 149(37.6) | 87(58.3) | 62(41.7) | |
| Schizophrenia | 103(26.0) | 61(59.2) | 42(40.1) | |
| Major depression | 66(16.7) | 31(47.0) | 35(53.0) | |
| Anxiety disorder | 14(3.5) | 7(50.0) | 7(50.0) | |

*(Continued)*

**Table 1.** (Continued)

| Characteristic | Overall N = 396 n (%) | Adherence, N = 213 n (%) | Non-adherence, N = 183 n (%) | P value |
|---|---|---|---|---|
| Schizo-affective disorder | 26(6.6) | 17(65.3) | 9(34.7) | |
| HIV Psychosis | 11(2.8) | 2(18.2) | 9(81.8) | |
| Alcohol psychosis | 15(3.8) | 3(20.0) | 12(80.0) | |
| Others | 12(3.1) | 5(41.6) | 7(58.4) | |
| **Duration of disease (yrs)** | | | | 0.268 |
| 1–5 | 282(71.2) | 144(51.1) | 138(48.9) | |
| 6–10 | 63(15.9) | 36(57.1) | 27(42.9) | |
| 11+ | 51(12.9) | 33(64.7) | 18(36.3) | |
| **Treatment duration (yrs)** | | | | 0.251 |
| 1–5 | 300(75.8) | 151(50.3) | 149(49.7) | |
| 6–10 | 54(13.6) | 35(64.8) | 19(35.2) | |
| 11+ | 42(10.6) | 27(64.2) | 15(35.8) | |
| **Pill number** | | | | 0.477 |
| <3 | 173(43.7) | 97(56.0) | 76(44.0) | |
| 3+ | 223(56.3) | 116(52.0) | 107(48.0) | |
| **Pill frequency** | | | | 0.195 |
| Once | 126(31.8) | 74(58.7) | 52(41.3) | |
| Twice | 269(67.9) | 139(51.7) | 130(48.3) | |
| Thrice | 1(0.3) | 0(0.0) | 1(100) | |
| **Side effects** | | | | **<0.001** |
| No | 275(69.4) | 165(60.0) | 110(40.0) | |
| Yes | 121(30.6) | 48(39.7) | 73(60.3) | |
| **Chronic disease** | | | | 0.032 |
| No | 347(87.6) | 194(55.9) | 153(44.1) | |
| Yes | 49(12.4) | 19(38.8) | 30(61.2) | |
| **Substance use** | | | | **<0.001** |
| No | 323(81.6) | 194(60.0) | 129(40.0) | |
| Yes | 73(18.4) | 19(26.0) | 54(74.0) | |

JRRH: Jinja Regional Referral Hospital; KIU-TH: Kampala International University Teaching Hospital; BAD: Bipolar affective disorder; HIV: Human Immunodeficiency Virus.

At the multivariate level of analysis, the variables that were independently associated with psychotropic medication non-adherence were poor family support, belief that mental illness is caused by witchcraft/sorcery, having side effects, and substance use. The rest of the details of the multivariate analysis are shown in Table 3 below.

### 3.4 Personality traits associated with psychotropic medication adherence among people with mental illness

In bivariate analysis, only neuroticism and agreeableness were the two personality traits with p- a p-value less than 0.2 and qualified for multivariate analysis (Table 4).

At multivariate analysis, after controlling for potential confounders which included the absence of family support, the belief that mental illness is caused by witchcraft/sorcery, having side effects, substance use, agreeableness, and neuroticism personality traits remained significantly associated with non-adherence. A participant with personality trait neuroticism was

**Table 2. Bivariate analysis of sociodemographic and clinical factors associated with psychotropic medication non-adherence among people with mental illness at KIU-TH and JRRH.**

| Characteristic | Adherence, N = 213 n (%) | Non-adherence, N = 183 n (%) | Bivariate analysis | | |
|---|---|---|---|---|---|
| | | | cOR | 95% CI | P-value |
| **Age category** | | | | | |
| 18–30 | 91(53.8) | 78(46.2) | Ref | | |
| 31–45 | 90(55.2) | 73(44.8) | 0.946 | 0.614–1.458 | 0.802 |
| 46–75 | 32(50.0) | 32(50.0) | 1.167 | 0.656–2.075 | 0.600 |
| **Sex** | | | | | |
| Male | 117(50.0) | 117(50.0) | 1.455 | 0.970–2.180 | **0.070** |
| Female | 96(59.2) | 66(40.8) | Ref | | |
| **Residence** | | | | | |
| Rural | 157(53.4) | 137(46.6) | 1.062 | 0.676–1.670 | 0.793 |
| Urban | 56(54.9) | 46(45.1) | Ref | | |
| **Education level** | | | | | |
| None | 19(55.8) | 15(44.2) | 1.911 | 0.807–4.528 | **0.141** |
| Primary | 44(40.4) | 65(59.6) | 3.577 | 1.853–6.902 | **0.000** |
| Secondary | 104(55.3) | 84(44.7) | 1.955 | 1.066–3.588 | **0.030** |
| Tertiary | 46(70.7) | 19(29.3) | Ref | | |
| **Employment** | | | | | |
| Employed | 51(64.5) | 28(35.5) | Ref | | |
| Self Employed | 64(46.0) | 75(54.0) | 2.134 | 1.208–3.771 | **0.009** |
| Unemployed | 98(55.0) | 80(45.0) | 1.487 | 0.860–2.571 | **0.156** |
| **Religion** | | | | | |
| Anglican | 68(52.3) | 62(47.7) | Ref | | |
| Catholic | 36(52.9) | 32(47.1) | 1.113 | 0.721–1.594 | 0.247 |
| Born again | 56(50.9) | 54(49.1) | 1.102 | 0.715–1.624 | 0.256 |
| Muslim | 48(57.8) | 35(42.2) | 1.204 | 0.738–1.965 | 0.277 |
| None | 2(40.0) | 3(60.0) | 0.220 | 0.024–1.986 | 0.235 |
| **Marital status** | | | | | |
| Married | 77(61.6) | 48(38.4) | Ref | | |
| Single | 103(53.9) | 88(46.1) | 1.371 | 0.866–2.169 | **0.179** |
| Separated/Divorced | 25(40.3) | 37(59.7) | 2.374 | 1.274–4.424 | **0.006** |
| Widow | 8(44.4) | 10(55.6) | 2.005 | 0.740–5.435 | **0.171** |
| **Family support** | | | | | |
| Poor | 26(23.4) | 85(76.6) | 6.238 | 3.774–10.313 | **<0.001** |
| Yes | 187(65.6) | 98(34.4) | Ref | | |
| **Hospital** | | | | | |
| JRRH | 178(54.9) | 146(45.1) | Ref | | |
| KIU-TH | 35(48.6) | 37(51.4) | 1.350 | 0.295–6.183 | 0.699 |
| **Beliefs about mental illness** | | | | | |
| Brain disorder | 159(62.3) | 96(37.7) | Ref | | |
| Witchcraft/Sorcery | 22(32.8) | 45(67.2) | 3.388 | 1.917–5.987 | **<0.001** |
| Demon possession | 19(50.0) | 19(50.0) | 1.656 | 0.835–3.284 | **0.149** |
| Ancestor's spirits | 13(36.1) | 23(63.9) | 2.930 | 1.418–6.055 | **0.004** |
| **Diagnosis** | | | | | |
| BAD | 87(58.3) | 62(41.7) | Ref | | |
| Schizophrenia | 61(59.2) | 42(40.8) | 0.966 | 0.580–1.610 | 0.895 |
| Major depression | 31(47.0) | 35(53.0) | 1.584 | 0.884–2.838 | 0.222 |

*(Continued)*

**Table 2.** (Continued)

| Characteristic | Adherence, N = 213 n (%) | Non-adherence, N = 183 n (%) | Bivariate analysis | | |
|---|---|---|---|---|---|
| | | | cOR | 95% CI | P-value |
| Anxiety disorder | 7(50.0) | 7(50.0) | 1.403 | 0.468–4.203 | 0.545 |
| Schizoaffective disorder | 17(65.3) | 9(34.7) | 0.743 | 0.311–1.775 | 0.504 |
| HIV Psychosis | 2(18.2) | 9(81.8) | 6.315 | 0.318–30.243 | 0.221 |
| Alcohol Psychosis | 3(20.0) | 12(80.0) | 5.613 | 0.520–20.728 | 0.210 |
| Others | 5(41.6) | 7(58.4) | 0.509 | 0.154–1.678 | 0.267 |
| **Duration of disease (yrs)** | | | | | |
| 1–5 | 144(51.1) | 138(48.9) | 1.757 | 0.945–3.266 | 0.275 |
| 6–10 | 36(57.1) | 27(42.9) | 1.375 | 0.643–2.942 | 0.412 |
| 11+ | 33(64.7) | 18(36.3) | Ref | | |
| **Treatment duration (yrs)** | | | | | |
| 1–5 | 151(50.3) | 149(49.7) | 1.776 | 0.908–3.473 | 0.293 |
| 6–10 | 35(64.8) | 19(35.2) | 0.977 | 0.421–2.270 | 0.957 |
| 11+ | 27(64.2) | 15(35.8) | Ref | | |
| **Pill number** | | | | | |
| <3 | 97(56.0) | 76(44.0) | Ref | | |
| 3+ | 116(52.0) | 107(48.0) | 1.177 | 0.790–1.754 | 0.423 |
| **Pill frequency** | | | | | |
| Once | 74(58.7) | 52(41.3) | Ref | | |
| Twice | 139(51.7) | 130(48.3) | 1.331 | 0.868–2.041 | **0.190** |
| Thrice | 0(0.0) | 1(100.0) | | | |
| **Side effects** | | | | | |
| No | 165(60.0) | 110(40.0) | Ref | | |
| Yes | 48(39.7) | 73(60.3) | 2.281 | 1.474–3.531 | **<0.001** |
| **Chronic disease** | | | | | |
| No | 194(55.9) | 153(44.1) | Ref | | |
| Yes | 19(38.8) | 30(61.2) | 2.002 | 1.085–3.694 | **0.026** |
| **Substance use** | | | | | |
| No | 194(60.0) | 129(40.0) | Ref | | |
| Yes | 19(26.0) | 54(74.0) | 4.274 | 2.421–7.545 | **<0.001** |

Ref = Reference category, cOR = Crude odds ratio, CI = Confidence interval.

7.424 times more likely to be non-adherent (aOR = 7.424, CI = 3.890–14.168, P<0.001). A participant with personality trait agreeableness was 0.062 times less likely to be non-adherent (aOR = 0.062, CI = 0.024–0.160, P<0.001). This means that a participant with personality trait agreeableness was 16.12 (1/0.062) times more likely to be adherent to medications (Table 5).

## 4. Discussion

In this hospital-based cross-sectional study we aimed to determine the prevalence of psychotropic medication non-adherence and associated personality traits (but also other associated factors or confounders) among people with mental illness.

In our study, the prevalence of psychotropic medication non-adherence was 46.21(95%CI 41.4–50.8). The high prevalence of medication non-adherence found in our study can also be explained by the false beliefs about mental illness in Uganda, among other factors.

Our prevalence was in a range of previous studies done in Nigeria and Malawi. In Nigeria, a study done by Adewuya has shown that 48% of the participants were non-adherent to

**Table 3. Multivariate analysis of sociodemographic and clinical factors associated with psychotropic medication non-adherence among people with mental illness at KIU-TH and JRRH.**

| Characteristic | Bivariate analysis | | | | Multivariate analysis | P value |
|---|---|---|---|---|---|---|
| | cOR | 95%CI | P value | aOR | 95%CI | |
| **Sex** | | | | | | |
| Male | 1.455 | 0.970–2.180 | 0.070 | 1.069 | 0.643–1.778 | 0.796 |
| Female | Ref | | | | | |
| **Education level** | | | | | | |
| None | 1.911 | 0.807–4.528 | 0.141 | 0.869 | 0.281–2.691 | 0.808 |
| Primary | 3.577 | 1.853–6.902 | <0.001 | 1.864 | 0.790–4.396 | 0.155 |
| Secondary | 1.955 | 1.066–3.588 | 0.030 | 1.296 | 0.600–2.801 | 0.509 |
| Tertiary | Ref | | | | | |
| **Employment** | | | | | | |
| Employed | Ref | | | | | |
| Self Employed | 2.134 | 1.208–3.771 | 0.009 | 1.412 | 0.684–2.913 | 0.350 |
| Unemployed | 1.487 | 0.860–2.571 | 0.156 | 0.942 | 0.468–1.893 | 0.866 |
| **Marital status** | | | | | | |
| Married | Ref | | | | | |
| Single | 1.371 | 0.866–2.169 | 0.179 | 0.712 | 0.401–1.263 | 0.245 |
| Separated | 2.374 | 1.274–4.424 | 0.006 | 0.721 | 0.315–1.648 | 0.438 |
| Widow | 2.005 | 0.740–5.435 | 0.171 | 0.568 | 0.157–2.064 | 0.390 |
| **Family support** | | | | | | |
| **Poor** | **6.238** | **3.774–10.313** | **<0.001** | **6.915** | **3.679–12.998** | **<0.001** |
| Good | Ref | | | | | |
| **Beliefs about mental illness** | | | | | | |
| Brain disorder | Ref | | | | | |
| **Witchcraft/Sorcery** | **3.388** | **1.917–5.987** | **<0.001** | **2.959** | **1.488–5.884** | **0.002** |
| Demon possession | 1.656 | 0.835–3.284 | 0.149 | 1.306 | 0.591–2.887 | 0.510 |
| Ancestor's spirit | 2.930 | 1.418–6.055 | 0.004 | 2.088 | 0.871–5.009 | 0.099 |
| **Pill frequency** | | | | | | |
| Once | Ref | | | | | |
| Twice | 1.331 | 0.868–2.041 | 0.190 | 0.885 | 0.514–1.523 | 0.658 |
| **Side effects** | | | | | | |
| No | Ref | | | | | |
| **Yes** | **2.281** | **1.474–3.531** | **<0.001** | **2.257** | **1.326–3.843** | **0.003** |
| **Chronic disease** | | | | | | |
| No | Ref | | | | | |
| Yes | 2.002 | 1.085–3.694 | 0.026 | 1.874 | 0.895–3.920 | 0.096 |
| **Substance use** | | | | | | |
| No | Ref | | | | | |
| **Yes** | **4.274** | **2.421–7.545** | **<0.001** | **4.174** | **2.121–8.214** | **<0.001** |

aOR = Adjusted odds ratio, Ref = Reference category, CI = Confidence interval

cOR = Crude odds ratio

medications [24] and Mekani in Malawi has found that 46.2% of the study participants were non-adherent to psychotropic medication [25]. However, the prevalence of non-adherence in our study was higher than the prevalence of medication non-adherence found by Gudeta (37.7%) and Mekuriwa (39.3%) in Ethiopia as well as 36.4% found by Ibrahim in Nigeria [12, 14, 26]. The prevalence was lower than 61.8% found by Nega et al. in Ethiopia and 58.7%

**Table 4. Bivariate analysis of traits associated with psychotropic medication non-adherence among people with mental illness at KIU-TH and JRRH.**

| Personality | Adherence, N = 213 | Non-adherence, N = 183 | Bivariate analysis | |
|---|---|---|---|---|
| | | | cOR(95%CI) | P value |
| **Extraversion** | | | | |
| Low | 178 (54.1) | 151(45.1) | Ref | |
| High | 35(52.2) | 32(47.8) | 1.078(0.637–1.824) | 0.780 |
| **Neuroticism** | | | | |
| Low | 200(66.9) | 99(33.1) | Ref | |
| **High** | **13(13.4)** | **84(86.6)** | **13.054(6.939–24.555)** | **<0.001** |
| **Openness to experience** | | | | |
| Low | 184(55.0) | 150(45.0) | Ref | |
| High | 29(55.7) | 33(44.3) | 1.396(0.811–2.404) | 0.229 |
| **Conscientiousness** | | | | |
| Low | 169(52.2) | 154(47.8) | Ref | |
| High | 44(60.3) | 29(39.7) | 0.723(0.431–1.213) | 0.220 |
| **Agreeableness** | | | | |
| Low | 121(40.5) | 178(59.5) | Ref | |
| **High** | **92(94.8)** | **5(5.2)** | **0.037(0.015–0.094)** | **<0.001** |

cOR = crude odds ratio, Ref = Reference category, CI = Confidence interval

**Table 5. Multivariate analysis of personality traits associated with psychotropic medication non-adherence among people with mental illness at KIU-TH and JRRH (controlling for potential confounders).**

| Characteristic | Bivariate analysis | | | Multivariate analysis | | |
|---|---|---|---|---|---|---|
| | cOR | 95%CI | P value | aOR | 95%CI | P value |
| **Neuroticism** | | | | | | |
| Low | Ref | | | | | |
| **High** | **13.054** | **6.939–24.555** | **<0.001** | **7.424** | **3.890–14.168** | **<0.001** |
| **Agreeableness** | | | | | | |
| Low | Ref | | | | | |
| **High** | **0.037** | **0.015–0.094** | **<0.001** | **0.062** | **0.024–0.160** | **<0.001** |
| **Family support** | | | | | | |
| No | **6.238** | **3.774–10.313** | **<0.001** | **6.915** | **3.679–12.998** | **<0.001** |
| Yes | Ref | | | | | |
| **Beliefs about mental illness** | | | | | | |
| Brain disorder | Ref | | | | | |
| Witchcraft/Sorcery | **3.388** | **1.917–5.987** | **<0.001** | **2.867** | **1.385–5.682** | **0.002** |
| Demon possession | 1.656 | 0.835–3.284 | 0.169 | 1.348 | 0.598–2.896 | 0.516 |
| Ancestor's spirit | 2.930 | 1.418–6.055 | 0.004 | 2.064 | 0.865–5.018 | 0.096 |
| **Side effects** | | | | | | |
| No | Ref | | | | | |
| Yes | **2.281** | **1.474–3.531** | **<0.001** | **2.242** | **1.322–3.838** | **0.001** |
| **Substance use** | | | | | | |
| No | Ref | | | | | |
| Yes | **4.274** | **2.421–7.545** | **<0.001** | **4.146** | **2.114–8.204** | **<0.001** |

aOR = Adjusted odds ratio, cOR = Crude odds ration Ref = Reference category, CI = Confidence interval.

found in India by Nagesh et al. [27, 28]. This difference could result from the difference in study settings but also in the tools used to assess medication adherence and personality traits.

## 4.1 Personality traits associated with psychotropic medication non-adherence

In our study, neuroticism and agreeableness were the two personality traits significantly associated with medication non-adherence after controlling for the significant sociodemographic and clinical factors.

In our study, participants with neuroticism traits were more likely to be non-adherent to psychotropic medications than other patients. Persons with predominant neuroticism personality traits are likely to have instability in emotions, self-doubt, irritability, anxiety, and depression. They experience generally negative emotions which can lead to poor adherence to medications. Our results were in line with other studies done in different parts of the world [20, 29]. However, a study done in Italy using the short version of the Big Five Inventory (BFI-10) to assess personality traits has found that conscientiousness was also associated with non-adherence to medication [19]. The difference can be explained by the small sample size (13 participants) and the longitudinal nature of the study.

Agreeableness was also significantly associated with psychotropic medication non-adherence. A participant with predominant agreeableness traits was less likely to be non-adherent to medications. The patients with predominant agreeableness personality traits tend to agree and follow other's opinions which can help to adhere to instructions given by health workers and to increase the level of adherence to medications.

This is similar to the results of a study done in Sweden which found that agreeableness influences positively adherence behavior [30]. In a systematic review done in India, agreeableness was also among the personality traits associated with adherence to medications in patients with chronic conditions [20].

However, in the USA, the most common personality traits associated with non-adherence among participants with bipolar affective disorder were extraversion, openness, and conscientiousness [31]. This difference was due to the study population (our study was done in patients with different mental disorders, not only in patients with bipolar affective disorder) and also the tool used in their study (the revised Five-Factor Inventory "Neo-FFI" with 60 items). In the U.K only conscientiousness was the personality trait influencing non-adherence behavior [32].

This difference may be explained by the different versions of the Big Five Inventory used to assess personality traits, the methodology used (longitudinal and multicenter), and also the sample size (1062 patients).

Other factors significantly associated with medication non-adherence in our study were poor family support, the belief that mental illness is caused by witchcraft/sorcery, the presence of side effects, and substance use.

Poor family support had a negative influence on medication adherence. In Uganda where there is a lack of social protection agencies, family is the principal source of support but many families in low-income countries have social and financial difficulties [33]. Family support improves medication adherence by taking care of the patient, reminding him to take medicine, and providing basic needs and money for transport to the hospital or to buy medications. The false beliefs about the causes of mental illness influenced psychotropic medication non-adherence. The beliefs about the causes of mental illness influence the choice to adhere to modern treatment [34] which leads to poor adherence to biological treatment. The false beliefs about the causes of mental illness lead to poor insight about illness and poor insight leads to psychotropic medication non-adherence [12]. The presence of side effects of medication was

significantly associated with medication non-adherence in our study. Individuals who experienced side effects were more likely to be non-adherent compared to those without side effects. Patients who experience severe side effects can intentionally stop medications to relieve the side effects. Substance use was also significantly associated with psychotropic medication non-adherence in our study. Patients who used psychoactive substances were more likely to be non-adherent compared to others. Psychoactive substances impair cognition and rational thinking and are also associated with increased adverse effects and interactions with psychotropic medications [35] which can increase non-adherence to psychotropic medications.

### 4.2 Strengths of the study

The study was done in public and private referral hospitals; the findings may apply to both public and private institutions. This is the first study exploring the association between personality traits and psychotropic medication non-adherence in Uganda and the East African region (to the best of our knowledge).

### 4.3 Limitations of the study

The study was cross-sectional and measured data at a single point in time. The study was done in only two hospitals chosen by convenience, so the results of the study cannot be generalized to the country. The questionnaire was administered to the patients but the responses were more subjective as they were self-reported. It has not been able to explore all possible factors associated with psychotropic medication non-adherence.

## 5. Conclusion

The results of this study show that medication non-adherence was high. Non-adherent patients were more likely to have predominant neuroticism personality traits. Medication non-adherence was shown to be less common in individuals with agreeableness personality traits. Other factors associated with medication non-adherence were poor social support, witchcraft/sorcery beliefs, side effects of medications, and substance abuse. Systematic assessment of personality traits should be done for all patients taking psychotropic medications. The findings from this study might help clinicians and other health caregivers to identify patients at high risk of being non-adherent who should need special care. This approach could help to offer reinforced psycho-education (after informed consent) for those patients and their caretakers. Shared decision-making and self-determination could help to increase adherence to medications. Community-based interventions for raising awareness about mental illnesses could help to decrease non-adherence to psychotropic medications. All stakeholders including tutors and trustees should be involved in those interventions. This study could give information to clinicians and policy-makers to develop personalized interventions. Further studies on health system-related factors are necessary to identify other factors associated with psychotropic medication non-adherence for global interventions.

## Supporting information

**S1 File. Dataset.**
(XLSX)

## Acknowledgments

The research assistant Waiswa Daniel (psychiatric clinical officer, JRRH) participated in data collection. We appreciate also all the patients who agreed to participate in the study.

## Author Contributions

**Conceptualization:** Emmanuel Niyokwizera.

**Data curation:** Emmanuel Niyokwizera, David Nitunga, Raissa Marie Ingrid Niyubahwe.

**Formal analysis:** Emmanuel Niyokwizera, Joshua Muhumuza, Joseph Kirabira.

**Investigation:** Emmanuel Niyokwizera.

**Methodology:** Emmanuel Niyokwizera.

**Supervision:** Emmanuel Niyokwizera, Nnaemeka Chukwudum Abamara, Joseph Kirabira.

**Validation:** Emmanuel Niyokwizera.

**Visualization:** Emmanuel Niyokwizera.

**Writing – original draft:** Emmanuel Niyokwizera.

**Writing – review & editing:** Emmanuel Niyokwizera, Nnaemeka Chukwudum Abamara, Joseph Kirabira.

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
