## [Decision Letter · Decision Letter 0]

4 Jul 2024

PONE-D-24-12886Personality traits and other factors associated with psychotropic medication non-adherence at two hospitals in Uganda. A cross-sectional study.PLOS ONE

Dear Dr. Niyokwizera,

Thank you for submitting your manuscript to PLOS ONE. After careful consideration, we feel that it has merit but does not fully meet PLOS ONE’s publication criteria as it currently stands. Therefore, we invite you to submit a revised version of the manuscript that addresses the points raised during the review process.

We look forward to receiving your revised manuscript.

Kind regards,

Roberto Scendoni

Academic Editor

PLOS ONE

Journal Requirements:

2. Thank you for submitting the above manuscript to PLOS ONE. During our internal evaluation of the manuscript, we found significant text overlap between your submission and previous work in the [introduction, conclusion, etc.].

Please revise the manuscript to rephrase the duplicated text, cite your sources, and provide details as to how the current manuscript advances on previous work. Please note that further consideration is dependent on the submission of a manuscript that addresses these concerns about the overlap in text with published work.

[If the overlap is with the authors’ own works: Moreover, upon submission, authors must confirm that the manuscript, or any related manuscript, is not currently under consideration or accepted elsewhere. If related work has been submitted to PLOS ONE or elsewhere, authors must include a copy with the submitted article. Reviewers will be asked to comment on the overlap between related submissions (http://journals.plos.org/plosone/s/submission-guidelines#loc-related-manuscripts).]

We will carefully review your manuscript upon resubmission and further consideration of the manuscript is dependent on the text overlap being addressed in full. Please ensure that your revision is thorough as failure to address the concerns to our satisfaction may result in your submission not being considered further.

3. We noted in your submission details that a portion of your manuscript may have been presented or published elsewhere. [No, except a preprint available on research square "Psychotropic medication adherence and associated personality traits in Uganda: a hospital-based cross-sectional study" but it is not under review in any journal and I have updated the article now with important changes before I submit the article to this journal.] Please clarify whether this [conference proceeding or publication] was peer-reviewed and formally published. If this work was previously peer-reviewed and published, in the cover letter please provide the reason that this work does not constitute dual publication and should be included in the current manuscript.

6. We are unable to open your Supporting Information file [Emma Data 10th November.sa]. Please kindly revise as necessary and re-upload.

Additional Editor Comments:

Have you encountered any limitations on the impossibility of administering the questionnaire? in what situations? Please indicate this within the limits by expanding this section.

Please add further conclusive starting points, such as the involvement of tutors or trustees to reduce the phenomenon of non-adherence to medication.

Rewrite the abstract based on the indications provided by the reviewers.

Reviewers' comments:

Reviewer's Responses to Questions

**Comments to the Author**

1. Is the manuscript technically sound, and do the data support the conclusions?

Reviewer #1: Yes

Reviewer #2: Partly

2. Has the statistical analysis been performed appropriately and rigorously? 

Reviewer #1: Yes

Reviewer #2: Yes

3. Have the authors made all data underlying the findings in their manuscript fully available?

Reviewer #1: Yes

Reviewer #2: Yes

4. Is the manuscript presented in an intelligible fashion and written in standard English?

Reviewer #1: Yes

Reviewer #2: Yes

5. Review Comments to the Author

Reviewer #1: The article addresses a current topic; however, there are suggestions that authors should follow to improve the quality of the manuscript.

-The psychosocial aspects of medication nonadherence have been little addressed. Are there any prejudices? What are the main factors that can aggravate or weaken a state of mental lability? Poverty? The fragility of an elderly person? The state of abandonment? An unexpected pregnancy? Please expand this discussion by adding further references (e.g: https://doi.org/10.1016/j.rcsop.2022.100178; https://doi.org/10.3390/healthcare11030328; https://doi.org/10.1111/bcp.14075).

-The conclusions are too limited and one topic that is poorly explored is that of the ethical question. For example, is imparting enhanced psychoeducation always correct or could it raise ethical issues, such as the lack of the principle of self-determination and informed consent? Please argue these aspects by referring to further literature (e.g: https://doi.org/10.3389/fpsyg.2022.935702; https://doi.org/10.3390/philosophies9030061; https://doi.org/10.1080/17522439.2023.2285967)

Reviewer #2: Overall, the study provides useful information that highlights non-adherence to antipsychotic medications in Uganda. The study methods and results generally provide adequate context. The focus on the level of psychotropic medication non-adherence and associated personality traits is a strength of the study. However, the authors should consider the following to strengthen it:

BACKGROUND:

The following statement seems problematic: "In Sub-Saharan African, by 2050, the burden due to psychiatric disorders and substances use disorders is estimated to increase globally by 130% and specifically by 139% in Eastern Africa

if there is no change in the prevalence and management of these disorders (2)." If Sub-Saharan Africa is the focus, is globally needed?" (page 2)

"Patients with psychiatric disorders are more likely to be non-adherent to treatment than those with

medical diseases (5)." (page 3). This statement makes it look like psychiatric disorders are not "medical diseases". Consider adding "other" before just "diseases".

The manuscript cites studies in India and Italy about how some personality traits affect non-adherence but similar studies in Africa were not cited. Did the authors not find a single study on antipsychotic medication non-adherence and personality traits in Africa to cite? If no such study exists, then the authors should consider indicating that as a gap in research for which their study aims to partly address. If there are similar studies in Africa too, the authors should consider citing them.

METHODS

The authors should consider indicating whether the survey was pilot-tested, and whether patients completed the questionnaire themselves (self-administered). If not self-administered, the authors should indicate whether those who administered received any training to minimise potential bias. How many people administered the questionnaires?

RESULTS

Repetition of "during the study period" in first sentence.

Table 1A: How is "Born Again" considered a religion? Did you mean "Pentecostal churches" or similar Charismatic churches?

Define abbreviations such as JRRH

TABLE 1B: Define BAD and HIV

TABLE 2: Consider defining cOR under "Bivariate Analysis" as has been done for aOR

DISCUSSION

In the first paragraph, the authors wrote: "The high prevalence of medication non-adherence found in our study can also be explained by the false beliefs about mental illness and the use of alternative medicine in Uganda, among others factors." However, the study did not assess use of alternative medicines in Uganda. Authors should consider citing at least a study showing that use of alternative medicine contributes to medication nonadherence otherwise they should consider not blaming alternative medicine for nonadherence.

Authors should consider assessing their in-text citation again. It seems references 19 and 20 in the text should be 20 and 21 in the reference list.

Authors should consider avoiding the terms "non-compliance" (pages 3 and 16) as non-adherence is the preferred term.

Authors claim their study is the first in Uganda and in East Africa on the subject (page 16). This is a bold statement.

As indicated earlier, the authors should consider examining the literature on antipsychotic medication nonadherence and personality traits again to determine similar studies done elsewhere in Africa. The can then discuss their findings in the context of the findings of those studies. If they cannot find any such study in Africa, they should consider adding "the best of our knowledge" to qualify that statement on page 16

The strengths and limitations are written like those of theses. Authors should consider rewriting them without bulleted points.

ALL OTHER SECTIONS

There are several issues with punctuation (eg, missing full stops; page 3 (after adding reference 8); page 7 after below") and inconsistencies that need to be addressed. For example "non adherence" (several pages) and "non-adherence" (page 3).

6. PLOS authors have the option to publish the peer review history of their article (what does this mean?). If published, this will include your full peer review and any attached files.

Reviewer #1: No

Reviewer #2: No

---

## [Author Response · Author response to Decision Letter 0]

12 Aug 2024

PONE-D-24-12886

Personality traits and other factors associated with psychotropic medication non-adherence at two hospitals in Uganda. A cross-sectional study.

PLOS ONE

Dear academic editor/PLOS ONE,

Thank you for considering my manuscript and the comments given to improve the work.

I have tried now to follow PLOS ONE's style requirements as it currently stands.

I would like to emphasize that there is no text overlap or duplication from published works or any related previous published works. As you can see using tools for plagiarism (Turnitin for example), the overlap is within the preprint of this manuscript sent to medrixv.org from PLOS ONE during the submission process and is now available on Cold Spring Harbor Laboratory. The preprint has never been published or presented elsewhere and is not under consideration elsewhere.

I would like to address the comments in form of a table which summarizes the comments and how the comments have been addressed.

I look forward for receiving positive feed back.

Kind regards,

Niyokwizera Emmanuel Principal investigator.

Summary of the comments How the comments have been addressed

Journal Requirements:

1. Please ensure that your manuscript meets PLOS ONE's style requirements, including those for file naming. We have adhered to style requirement now.

2.During our internal evaluation of the manuscript, we found significant text overlap between your submission and previous work in the [introduction, conclusion, etc.]. Please revise the manuscript to rephrase the duplicated text, cite your sources, and provide details as to how the current manuscript advances on previous work.

If the overlap is with the authors’ own works: Moreover, upon submission, authors must confirm that the manuscript, or any related manuscript, is not currently under consideration or accepted elsewhere. The overlap is within the preprint of this manuscript sent to medrixv.org from PLOS ONE during the initial submission process to PLOS ONE and available now on Cold Spring Harbor Laboratory, partner to PLOS ONE. 

There is no extract or copy from previous presented or published works. 

This manuscript or his preprint has not been published and is not under consideration elsewhere. No related manuscript is under consideration or has been

3. We noted in your submission details that a portion of your manuscript may have been presented or published elsewhere. [No, except a preprint available on research square ". Please clarify whether this [conference proceeding or publication] was peer-reviewed and formally published. That related preprint was not presented in any conference, was not peer-reviewed or accepted or published in any journal or elsewhere. 

4. Your ethics statement should only appear in the Methods section of your manuscript. If your ethics statement is written in any section besides the Methods, please delete it from any other section. Noted and it has been deleted from other sections.

5. Please include captions for your Supporting Information files at the end of your manuscript, and update any in-text citations to match accordingly. It has been included.

6. We are unable to open your Supporting Information file [Emma Data 10th November.sa]. Please kindly revise as necessary and re-upload. It has been revised to XLSL document and re-uploaded.

Additional Editor Comments:

Have you encountered any limitations on the impossibility of administering the questionnaire? in what situations? Please indicate this within the limits by expanding this section.

Please add further conclusive starting points, such as the involvement of tutors or trustees to reduce the phenomenon of non-adherence to medication.

Rewrite the abstract based on the indications provided by the reviewers. This was an error. The questionnaire was administered by the investigators and research assistant but the responses were more subjective as they were self-report from the patients.

It has been added in conclusion

It has been done

Review Comments to the Author

Reviewer #1: The article addresses a current topic; however, there are suggestions that authors should follow to improve the quality of the manuscript.

-The psycho-social aspects of medication non-adherence have been little addressed. Are there any prejudices? What are the main factors that can aggravate or weaken a state of mental lability? Poverty? The fragility of an elderly person? The state of abandonment? An unexpected pregnancy? Please expand this discussion by adding further references. 

The psycho-social aspect of medication non-adherence has been improved and further references have been added in accordance with these comments.

However, in our study findings, there was no significant association with non-adherence for the additional psycho-social factors mentioned in the comments. 

-The conclusions are too limited and one topic that is poorly explored is that of the ethical question. For example, is imparting enhanced psychoeducation always correct or could it raise ethical issues, such as the lack of the principle of self-determination and informed consent? Please argue these aspects by referring to further literature The conclusion has been revised.

The psycho-education is generally done after informed consent is obtained. When well done, the psycho-education can increase motivation and self-determination to change, including adherence to medications.

Reviewer #2: Overall, the study provides useful information that highlights non-adherence to antipsychotic medications in Uganda. The study methods and results generally provide adequate context. The focus on the level of psychotropic medication non-adherence and associated personality traits is a strength of the study. However, the authors should consider the following to strengthen it:

BACKGROUND:

The following statement seems problematic: "In Sub-Saharan African, by 2050, the burden due to psychiatric disorders and substances use disorders is estimated to increase globally by 130% and specifically by 139% in Eastern Africa

if there is no change in the prevalence and management of these disorders (2)." If Sub-Saharan Africa is the focus, is globally needed?" (page 2)

"Patients with psychiatric disorders are more likely to be non-adherent to treatment than those with

medical diseases (5)." (page 3). This statement makes it look like psychiatric disorders are not "medical diseases". Consider adding "other" before just "diseases".

The manuscript cites studies in India and Italy about how some personality traits affect non-adherence but similar studies in Africa were not cited. Did the authors not find a single study on antipsychotic medication non-adherence and personality traits in Africa to cite? If no such study exists, then the authors should consider indicating that as a gap in research for which their study aims to partly address. If there are similar studies in Africa too, the authors should consider citing them. 

Noted and corrected. Globally has been removed.

Noted and corrected

To the best of our knowledge, no studies on psychotropic medication non-adherence and personality traits in Africa. There is a gap in research in that area.

METHODS

The authors should consider indicating whether the survey was pilot-tested, and whether patients completed the questionnaire themselves (self-administered). If not self-administered, the authors should indicate whether those who administered received any training to minimize potential bias. How many people administered the questionnaires? The study was pilot-tested

The questionnaire was administered by 3 psychiatrist and 1 psychiatric clinical officer. They were trained before and received ongoing training during data collection.

RESULTS

Repetition of "during the study period" in first sentence.

Table 1A: How is "Born Again" considered a religion? Did you mean "Pentecostal churches" or similar Charismatic churches?

Define abbreviations such as JRRH

TABLE 1B: Define BAD and HIV

TABLE 2: Consider defining cOR under "Bivariate Analysis" as has been done for aOR 

Noted and corrected

It has been changed to Pentecostal churches.

DISCUSSION

In the first paragraph, the authors wrote: "The high prevalence of medication non-adherence found in our study can also be explained by the false beliefs about mental illness and the use of alternative medicine in Uganda, among others factors." However, the study did not assess use of alternative medicines in Uganda. Authors should consider citing at least a study showing that use of alternative medicine contributes to medication nonadherence otherwise they should consider not blaming alternative medicine for nonadherence.

Authors should consider assessing their in-text citation again. It seems references 19 and 20 in the text should be 20 and 21 in the reference list.

Authors should consider avoiding the terms "non-compliance" (pages 3 and 16) as non-adherence is the preferred term.

Authors claim their study is the first in Uganda and in East Africa on the subject (page 16). This is a bold statement.

As indicated earlier, the authors should consider examining the literature on antipsychotic medication non-adherence and personality traits again to determine similar studies done elsewhere in Africa. The can then discuss their findings in the context of the findings of those studies. If they cannot find any such study in Africa, they should consider adding "the best of our knowledge" to qualify that statement on page 16

The strengths and limitations are written like those of theses. Authors should consider rewriting them without bulleted points. 

Noted and corrected

Noted and corrected

Noted and corrected

Noted and corrected

Noted and corrected

Noted and corrected

ALL OTHER SECTIONS

There are several issues with punctuation (eg, missing full stops; page 3 (after adding reference 8); page 7 after below") and inconsistencies that need to be addressed. For example, "non adherence" (several pages) and "non-adherence" (page 3). 

Noted and corrected

---

## [Decision Letter · Decision Letter 1]

28 Aug 2024

PONE-D-24-12886R1Personality traits and other factors associated with psychotropic medication non-adherence at two hospitals in Uganda. A cross-sectional study.PLOS ONE

Dear Dr. Niyokwizera,

Thank you for submitting your manuscript to PLOS ONE. After careful consideration, we feel that it has merit but does not fully meet PLOS ONE’s publication criteria as it currently stands. Therefore, we invite you to submit a revised version of the manuscript that addresses the points raised during the review process.

Please follow the reviewers' suggestions to make the article complete, also comparing it with other scientific works on the topic. 

We look forward to receiving your revised manuscript.

Kind regards,

Roberto Scendoni

Academic Editor

PLOS ONE

Journal Requirements:

Reviewers' comments:

Reviewer's Responses to Questions

**Comments to the Author**

1. If the authors have adequately addressed your comments raised in a previous round of review and you feel that this manuscript is now acceptable for publication, you may indicate that here to bypass the “Comments to the Author” section, enter your conflict of interest statement in the “Confidential to Editor” section, and submit your "Accept" recommendation.

Reviewer #1: (No Response)

Reviewer #2: (No Response)

2. Is the manuscript technically sound, and do the data support the conclusions?

Reviewer #1: Yes

Reviewer #2: Yes

3. Has the statistical analysis been performed appropriately and rigorously? 

Reviewer #1: Yes

Reviewer #2: Yes

4. Have the authors made all data underlying the findings in their manuscript fully available?

Reviewer #1: Yes

Reviewer #2: (No Response)

5. Is the manuscript presented in an intelligible fashion and written in standard English?

Reviewer #1: Yes

Reviewer #2: No

6. Review Comments to the Author

Reviewer #1: The authors have considered the issues raised during the review. However, the Conclusions section continues to be sparse. Therefore, they are again urged to expand it by referring to additional literature (e.g., https://doi.org/10.3389/fpsyg.2022.935702; https://doi.org/10.3390/philosophies9030061; https://doi.org/10.1080/17522439.2023.2285967).

Reviewer #2: The authors have to a great extent addressed the issues I raised. However, this manuscript has serious language issues. I am unable to point all of them here.

Just consider the sentence below

"This is similar to the results of a study done in Sweden who found that agreeableness influences positively the adherence behavior [30] and a systematic review done in India, studies showed that Agreeableness was among the personality traits negatively associated with non-adherence for patients with chronic conditions[20]." (See discussion). The use of "who" is a problem, and the subsequent phrase beginning with "studies" is also a problem.

"Sociodemographic informations" appears four times but there should be no "s" after "information".

Even under the Aknowledgements, punctuation is a problem (no full stop after second sentence). Under Authors' contributions, the authors have not used proper tenses. They should consult recently published papers in PLOS to learn how to craft Authors contributions.

Authors should consider getting the help of someone who can adequately address such language issues if the authors cannot address them themselves.

7. PLOS authors have the option to publish the peer review history of their article (what does this mean?). If published, this will include your full peer review and any attached files.

Reviewer #1: No

Reviewer #2: No

---

## [Author Response · Author response to Decision Letter 1]

6 Sep 2024

PONE-D-24-12886R1

Personality traits and other factors associated with psychotropic medication non-adherence at two hospitals in Uganda. A cross-sectional study.

PLOS ONE

Dear academic editor,

Thank you for considering my manuscript and previous revisions.

I would like to address the additional comments in form of a table which summarizes the comments and how the comments have been addressed.

I look forward for receiving positive feed back.

Kind regards.

Comment How the comment has been addressed

Journal Requirements:

Please review your reference list to ensure that it is complete and correct. If you have cited papers that have been retracted, please include the rationale for doing so in the manuscript text, or remove these references and replace them with relevant current references. 

We have reviewed our reference list. It was complete. However, references 19, 25 and 29 were preprints not yet peer reviewed. We have found now their corresponding updated published articles. We have replaced them by the current references.

Reviewer #1: The authors have considered the issues raised during the review. However, the Conclusions section continues to be sparse. Therefore, they are again urged to expand it by referring to additional literature The conclusion section has been expanded by referring to additional literature.

Reviewer #2: The authors have to a great extent addressed the issues I raised. However, this manuscript has serious language issues. I am unable to point all of them here. 

Under Authors' contributions, the authors have not used proper tenses. They should consult recently published papers in PLOS to learn how to craft Authors contributions. The English language issues have been addressed with the help of colleagues and English specialists. All the mistakes have been corrected. 

The authors’ contribution section has been revised according to PLOS ONE format.

---

## [Decision Letter · Decision Letter 2]

22 Oct 2024

PONE-D-24-12886R2Personality traits and other factors associated with psychotropic medication non-adherence at two hospitals in Uganda. A cross-sectional study.PLOS ONE

Dear Dr. Niyokwizera,

Thank you for submitting your manuscript to PLOS ONE. After careful consideration, we feel that it has merit but does not fully meet PLOS ONE’s publication criteria as it currently stands. Therefore, we invite you to submit a revised version of the manuscript that addresses the points raised during the review process.

We look forward to receiving your revised manuscript.

Kind regards,

Roberto Scendoni

Academic Editor

PLOS ONE

Journal Requirements:

Reviewers' comments:

Reviewer's Responses to Questions

**Comments to the Author**

1. If the authors have adequately addressed your comments raised in a previous round of review and you feel that this manuscript is now acceptable for publication, you may indicate that here to bypass the “Comments to the Author” section, enter your conflict of interest statement in the “Confidential to Editor” section, and submit your "Accept" recommendation.

Reviewer #1: All comments have been addressed

Reviewer #2: (No Response)

2. Is the manuscript technically sound, and do the data support the conclusions?

Reviewer #1: (No Response)

Reviewer #2: Yes

3. Has the statistical analysis been performed appropriately and rigorously? 

Reviewer #1: (No Response)

Reviewer #2: Yes

4. Have the authors made all data underlying the findings in their manuscript fully available?

Reviewer #1: (No Response)

Reviewer #2: (No Response)

5. Is the manuscript presented in an intelligible fashion and written in standard English?

Reviewer #1: (No Response)

Reviewer #2: No

6. Review Comments to the Author

Reviewer #1: (No Response)

Reviewer #2: The authors have slightly improved the manuscript. However, the language issues have not been fully addressed.

I am unable to point all of them here, but the authors should consider the following:

INTRODUCTION

1. "The 2017 report of WHO indicated that globally, 80% of people with mental disorders are living in low- and middle-income countries (LMIC) which include Uganda and, up to 75% of affected persons in many LMIC did not have access to the needed treatment [3]."

SUGGESTION

"are" after "disorders" should be "were", LMIC should be "LMICs", "and, up to" should be ..., and up to" [comma before "and"

"Psycho-social and demographic factors (age, gender, marital status, occupation, family and social support, etc.), substance abuse, beliefs about mental illness are important factors associated with non-adherence related to the patient [5]."

SUGGESTION

Consider adding "and" before "beliefs".

"Illness related factors like side effects of medications, beliefs about the illness, treatments complexity and lack of insight are barriers to medication adherence [16]."

SUGGESTION

"Illness related factors" should be "Illness-related factors [hyphen between illness and related"

The manuscript would need additional editing for issues such as grammar, punctuation and tenses to make its presentation more intelligible.

7. PLOS authors have the option to publish the peer review history of their article (what does this mean?). If published, this will include your full peer review and any attached files.

Reviewer #1: No

Reviewer #2: No

---

## [Author Response · Author response to Decision Letter 2]

4 Nov 2024

PONE-D-24-12886R1

Personality traits and other factors associated with psychotropic medication non-adherence at two hospitals in Uganda. A cross-sectional study.

PLOS ONE

Dear academic editor,

Thank you for considering my manuscript and previous revisions.

I would like to address the additional comments in form of a table which summarizes the comments and how the comments have been addressed.

I look forward for receiving positive feed back.

Kind regards.

Comment How the comment has been addressed

Journal Requirements:

No changes found in reference list.

Reviewer #1: All comments have been addressed OK.

Reviewer #2: The authors have slightly improved the manuscript. However, the language issues have not been fully addressed.

I am unable to point all of them here, but the authors should consider the following:

1. "The 2017 report of WHO indicated that globally, 80% of people with mental disorders are living in low- and middle-income countries (LMIC) which include Uganda and, up to 75% of affected persons in many LMIC did not have access to the needed treatment [3]."

SUGGESTION

"are" after "disorders" should be "were", LMIC should be "LMICs", "and, up to" should be ..., and up to" [comma before "and"

"Psycho-social and demographic factors (age, gender, marital status, occupation, family and social support, etc.), substance abuse, beliefs about mental illness are important factors associated with non-adherence related to the patient [5]."

SUGGESTION

Consider adding "and" before "beliefs".

"Illness related factors like side effects of medications, beliefs about the illness, treatments complexity and lack of insight are barriers to medication adherence [16]."

SUGGESTION

"Illness related factors" should be "Illness-related factors [hyphen between illness and related"

The manuscript would need additional editing for issues such as grammar, punctuation and tenses to make its presentation more intelligible. The Language issue has been addressed.

Corrected.

Corrected.

Corrected.

Corrected.

The manuscript has been fully edited and all errors in grammar, punctuation, and tenses has been addressed.

---

## [Editor Report · Decision Letter 3]

11 Nov 2024

Personality traits and other factors associated with psychotropic medication non-adherence at two hospitals in Uganda. A cross-sectional study.

PONE-D-24-12886R3

Dear Dr. Niyokwizera,

We’re pleased to inform you that your manuscript has been judged scientifically suitable for publication and will be formally accepted for publication once it meets all outstanding technical requirements.

Kind regards,

Roberto Scendoni

Academic Editor

PLOS ONE
---

## [Editor Report · Acceptance letter]

13 Nov 2024

PONE-D-24-12886R3 

PLOS ONE

Dear Dr. Niyokwizera, 

I'm pleased to inform you that your manuscript has been deemed suitable for publication in PLOS ONE. Congratulations! Your manuscript is now being handed over to our production team.

Kind regards, 

on behalf of

Dr. Roberto Scendoni 

Academic Editor

PLOS ONE